

# Distinct role of Klotho in long bone and craniofacial bone: skeletal development, repair and regeneration

Xinyu Chen[1],*, Yali Wei[2],*, Zucen Li[2], Chenchen Zhou[3] and Yi Fan[2]

[1] State Key Laboratory of Oral Diseases & National Center for Stomatology & National Clinical Research Center for Oral Diseases, West China Hospital of Stomatology, Sichuan University, Chengdu, China

[2] State Key Laboratory of Oral Diseases & National Center for Stomatology & National Clinical Research Center for Oral Diseases & Department of Operative Dentistry and Endodontics, West China Hospital of Stomatology, Sichuan University, Chengdu, China

[3] State Key Laboratory of Oral Diseases & National Center for Stomatology & National Clinical Research Center for Oral Diseases & Department of Pediatric Dentistry, West China Hospital of Stomatology, Sichuan University, Chengdu, China

* These authors contributed equally to this work.

Corresponding authors
Chenchen Zhou,
chenchenzhou5510@scu.edu.cn
Yi Fan, yifan@scu.edu.cn

## ABSTRACT

Bone defects are highly prevalent diseases caused by trauma, tumors, inflammation, congenital malformations and endocrine abnormalities. Ideally effective and side effect free approach to dealing with bone defects remains a clinical conundrum. Klotho is an important protein, which plays an essential role in regulating aging and mineral ion homeostasis. More recently, research revealed the function of Klotho in regulating skeleton development and regeneration. Klotho has been identified in mesenchymal stem cells, osteoblasts, osteocytes and osteoclasts in different skeleton regions. The specific function and regulatory mechanisms of Klotho in long bone and craniofacial bone vary due to their different embryonic development, ossification and cell types, which remain unclear and without conclusion. Moreover, studies have confirmed that Klotho is a multifunctional protein that can inhibit inflammation, resist cancer and regulate the endocrine system, which may further accentuate the potential of Klotho to be the ideal molecule in inducing bone restoration clinically. Besides, as an endogenous protein, Klotho has a promising potential for clinical therapy without side effects. In the current review, we summarized the specific function of Klotho in long bone and craniofacial skeleton from phenotype to cellular alternation and signaling pathway. Moreover, we illustrated the possible future clinical application for Klotho. Further research on Klotho might help to solve the existing clinical difficulties in bone healing and increase the life quality of patients with bone injury and the elderly.

## INTRODUCTION

Klotho (KL) is divided into three subfamilies, α-Klotho, β-Klotho and γ-Klotho respectively. Full-length α-Klotho is a membrane-anchored protein with extracellular,

transmembrane and intracellular domains (*Kuro-o et al., 1997*; *Shiraki-Iida et al., 1998*). The extracellular domain of Klotho consists of two repeated sequences assigned KL1 and KL2 respectively. Between the repeats is a four basic amino acid sequence: Lys-Lys-Arg-Lys, which might be a potential proteolytic cleavage site (*Buendía et al., 2016*). *Chen et al. (2022b)* recently synthesized 18 overlapping peptides using GenScript based on the structure of KL1, namely Klotho-derived peptide 1 to 18 (KP1 to KP18), each encompassing 30 amino acids with some demonstrating biological function. α-Klotho is primarily expressed in the kidney, parathyroid gland and choroid plexus. Recent research confirmed its expression in other tissues such as bone, skin and anterior pituitary (*Olauson et al., 2017*). The gene encodes Klotho can be transcribed into two forms, individually membrane-anchored Klotho (membrane Klotho, mKL) and soluble Klotho (sKL). mKL is the full-length Klotho while sKL lacks the second repeated extracellular domain as well as the transmembrane and intracellular domain (*Matsumura et al., 1998*). The extracellular domain of Klotho can be cleaved by ADAM10 and ADAM17 and turn into sKL (*Chen et al., 2007*). By getting into circulation, sKL can be detected in blood, urine as well as cerebrospinal fluid and act as a systematically beneficial hormone (*Buendía et al., 2016*). After the revelation of α-Klotho, *Ito et al. (2000, 2002)* identified β-Klotho and γ-Klotho, which exhibit similar structure and biological functions to α-Klotho. We illustrated the structure of Klotho subfamilies in Fig. 1. Since α-Klotho is the earliest discovered and most studied Klotho protein in bone biology, we focused on α-Klotho in the current review.

First discovered in 1997, the protein encoded by *Klotho* was confirmed to related to aging (*Kuro-o et al., 1997*). *Klotho* deficient mice exhibited human aging manifestations, for instance, atrophy of the skin, genital organs, thymus and osteoporosis (*Kuro-o et al., 1997*). In contrast, mice overexpressing Klotho had insulin and insulin-like growth factor 1 (IGF-1) resistant, which might contribute to a longer life span (*Kuro-o et al., 1997*; *Kurosu et al., 2005*). The general function of Klotho in the kidney is to bind fibroblast growth factor receptor1 (FGFR1) and improve the affinity between fibroblast growth factor 23 (FGF-23) and FGFR1 to participate in phosphorus metabolism and regulate vitamin D homeostasis (*Christov & Jüppner, 2018*). Moreover, Klotho is tightly associated with the health of multiple systems, such as the urinary system, cardiovascular system, neurological system, respiratory system as well as the physiological functions including maintaining mineral homeostasis and regulating bone metabolism (*Kuro-o, 2006*; *Prud'homme, Kurt & Wang, 2022*). *Rhee et al. (2011)* first reported Klotho expression in osteoblastic cells. In wild-type (WT) mice, *Klotho* mRNA level in the entire bone and in the separated osteoblasts was about 5 to 10-fold more than that in $Klotho^{-/-}$ mice (*Yuan et al., 2012*). *Raimann et al. (2013)* further verified the existence of Klotho protein in bone-forming cells and growth plate. Afterward, role of Klotho on bone independent of mineral metabolism has been extensively studied, focusing on how Klotho directly influences osteogenic differentiation of stem cells, osteogenesis and osteoclastogenesis. More recently, we discovered a specific role of Klotho in craniofacial bone which demonstrated a contrasting regulatory role compared to long bone and is essential for future clinical therapy in bone repair (*Fan et al., 2022*).

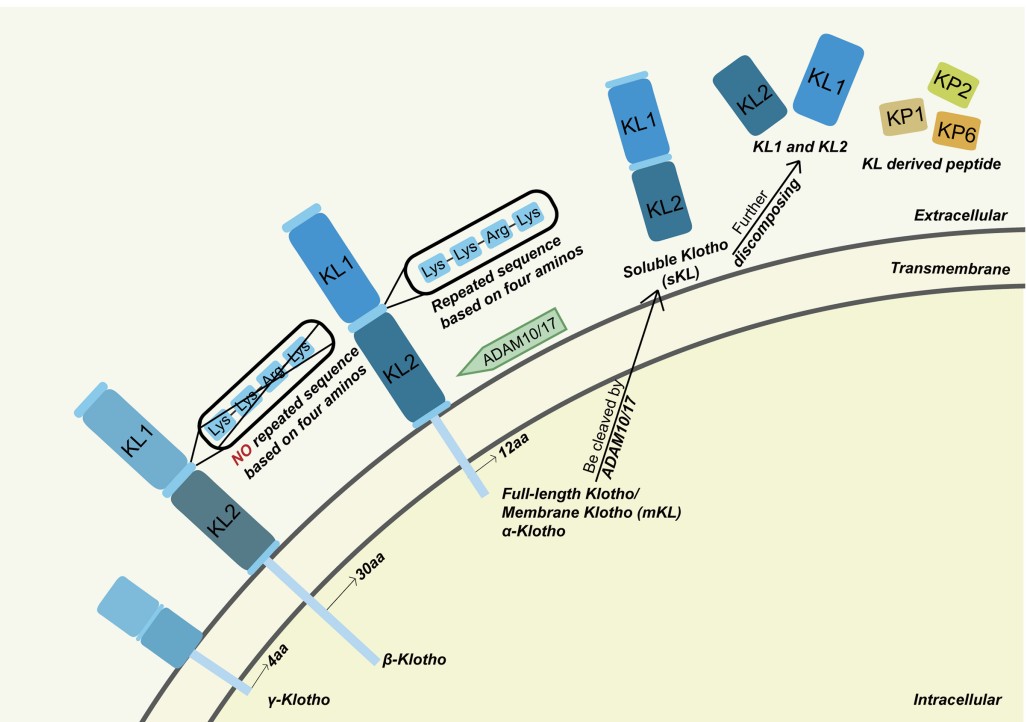

**Figure 1 The structure of Klotho.** Full-length α-Klotho is a transmembrane protein that can be cleaved into soluble Klotho which could then be discomposed into KL1 and KL2. The structure of β-Klotho shared large similarities with α-Klotho which contains three domains, respectively, extracellular, transmembrane and intracellular as well. β-Klotho may proteolyze into βKL1 and βKL2. However, no repeated amino acid sequences are located between βKL1 and βKL2. For γ-Klotho, the overall structure resembles α-Klotho and β-Klotho, yet the full length is shorter. aa, acid amino; KP1, Klotho-derived peptide 1; KP2, Klotho-derived peptide 2; KP6, Klotho-derived peptide 6. Figure created using Adobe Illustrator.

Bone defects are highly prevalent nowadays, which might result from traumas, space-occupying lesions, endocrine disease, osteoporosis as well as complications such as inflammation and congenital etiologies. The aftermath of critical bone defects includes nonunion, malunion and fracture (*Aldhaher et al., 2023*). Bone defects in the maxillofacial region remain a clinical conundrum. Periodontitis or apical periodontitis (AP) are the general causes of orofacial bone loss in clinical practice. Studies illustrated that severe periodontitis featured by alveolar bone destruction affects more than 7.8% of adults in the U.S. and more than 52% of adults have at least one tooth suffering from AP (*Tibúrcio-Machado et al., 2021*). In addition, severe trauma or tumor may lead to extensive bone defects. At present, bone tissue engineering is committing to discovering a possible way that fulfills three requirements, respectively, filling lost bone tissues, adapting to changes in the body and no extra surgery for bone collection. Scaffolds with stem cells, cytokines or drugs to improve bone formation are widely investigated (*Laird et al., 2021*). Ideal substances might synergize multiple aspects to promote bone repair, including directly inducing osteogenesis, improving microenvironments such as reducing inflammation and controlling the progression of the primary diseases such as anti-tumor or anti-bacteria with the least adverse reaction (*Smolinska et al., 2023*).

As an endogenous protein, Klotho may hardly cause immune reactions or body rejection (*Shan et al., 2022*). In fact, for the past few years, Klotho has been proven to be effective in treating several diseases. For cardiac health, recombinant rat Klotho could reverse the heart damage caused by oxidative stress (*Olejnik et al., 2023*). In the kidney, injecting recombinant soluble Klotho could ameliorate diabetic nephropathy in mice (*Lee et al., 2021*). Recombinant human Klotho could suppress acute kidney injury by alleviating inflammation and oxidative stress. Notably, excessive Klotho may lead to an imbalance of phosphate homeostasis and result in terrible consequences. In addition, as a macromolecular, mass production for the treatment usage of Klotho is financially difficult. However, Klotho-derived peptide may solve those drawbacks. KP1 is effective in treating kidney fibrosis and KP6 has a significant role in treating diabetic kidney diseases (*Yuan et al., 2022*; *Chen et al., 2022b*). All the research suggests a potential therapeutic function for Klotho.

Although Klotho exhibited a promising role in repairing bone defects, its precise mechanism for regulating the skeleton remains unclear and lacks an integrate summary. In this review, we concluded the phenotype alternation in long bone and craniofacial bone as well as the Klotho-regulated cellular alternation and mechanism in bone metabolism. Finally, we put forward some suggestions concerning the possible therapeutic applications of Klotho in curing bone diseases and that might be useful for future research directions.

## SURVEY METHODOLOGY

The search of article was carried out in PubMed, Web of Science and Google Scholar using specific keywords or their combination including "Klotho," "bone," "long bone," "craniofacial bone," "bone defects," "regeneration," "stem cells," "osteoblasts," "osteoclasts," and "osteocytes." Researches on Klotho structure, bone formation and regeneration under the regulation of Klotho as well as the current therapeutic application of Klotho were included.

## THE RATIONALE FOR WHY IT IS NEEDED

Bone defect is a highly prevalent disease worldwide. The current therapeutic method for bone defects, which is bone grafting, has limitations. Artificial and xenogenic bone grafting may cause immune and inflammatory responses, leading to irreversible damage to the body. Although autologous bone does not cause such side effects, its obtainment requires additional injury. Currently, bone tissue engineering is committing to discovering novel substances that can treat the primary disease causing bone defects and promote bone regeneration without affecting physiological functions. Klotho is an essential protein in regulating bone development, remodeling and regeneration. It has distinct functions in various cells in long bone and craniofacial bone under normal and inflammatory microenvironment. As an endogenous protein with multiple functions, Klotho holds promise for clinical applications in promoting bone regeneration. However, there still exists insufficiency in concluding the distinct role of Klotho in bones.

In the current review, we aimed to provide an overview of recent advancements in the specific role of Klotho in bone development and remodeling focusing on long bone and

craniofacial bone. We characterized how Klotho manages the long bone and craniofacial skeleton from phenotype, cellular alternation to signaling pathway and highlighted the promising potential of Klotho as an endogenous protein for the clinical therapy targeting bone defects.

## THE AUDIENCE IT IS INTENDED FOR

Doctors specializing in orthopedics or dentistry as well as researchers majoring in bone repair and regeneration may be interested in this review. Studies that focus on the role of Klotho in the skeletal system offer an alternative for bone restoration. With continuous research and a developing understanding of the mechanisms of Klotho in tissue reconstruction, it might eventually be applied in the clinical treatment for patients suffering from bone defects.

### Phenotype alternation related to Klotho in long bone and craniofacial bone

Klotho was first found to be expressed in mice tibia as well as isolated osteoblasts and osteoclasts. Further research confirmed its expression in cortical bone and growth plate (*Raimann et al., 2013*; *Rhee et al., 2011*). Klotho-specific ablated mice exhibit distinct bone phenotypes, yet some of them are controversial. Besides, unlike most appendicular bones, the craniofacial skeleton originates from the neuroectoderm rather than the mesoderm. They are formed by intramembranous and endochondral ossification while long bone undergoes endochondral ossification (*Dong et al., 2022*; *Murali et al., 2016*). Craniofacial bone repair is more challenging due to its anatomical structure, complex morphology and physiological characteristics. In addition, in the mandibular bone, the development depends on the pressure gradient created by the attachment of the masticatory muscle. Once intramembranous bone formation is completed, three types of secondary cartilages, namely midline synchondrosis, angular synchondrosis, and condylar cartilage, will be formed as primary growth centers during the prenatal period. After birth, these secondary cartilages ossify or transform into secondary growth sites, which influences mandibular growth (*Roberts & Hartsfield, 2004*). Additionally, the regeneration of craniofacial bone goes through membranous ossification while long bone proceeds with endochondral ossification (*Tsou et al., 2023*). These may lead to a significant discrepancy in osteogenic potential, homeostatic properties, and receptor expressions in the craniofacial bone compared to other skeleton (*Matsuura et al., 2014*). Moreover, the maxillofacial skeleton has exceptional structure and function. Craniofacial bone loss may cause serious physical and mental problems to patients. For example, physiological dysfunction including breathing, swallowing and speaking (*Tan et al., 2021*). Thus, bone defects in the maxillofacial complex need more attention and are worthy of consideration individually.

Furthermore, inflammation and bone loss are usually connected. Infection from the tooth, periodontal tissues or blood resulting in local inflammation and leading to bone defects. In general, the inflammatory process will aggravate bone damage and mitigate the process of bone healing. Acute inflammation induces the production of chemokines to accelerate bone formation by promoting the osteoblastic differentiation and proliferation

of mesenchymal progenitor cells. In contrast, chronic inflammation may inhibit the synthesis of the promoting factors in new bone formation, thus promoting osteolytic lesion formation during periodontitis or AP (*Yang et al., 2021*). In this part, we concluded the role of Klotho in long bone and craniofacial bone respectively and discussed the impact of Klotho on craniofacial bone under inflammatory condition.

## Long bone

### Alterations of long bone in Klotho-related transgenic mouse models

The original discovery established a hypomorphic *kl/kl* mice through pronuclear microinjection. Those mice demonstrated human age-related osteoporosis, specifically, decreased bone mineral density (BMD) particularly evident in long bone, reduced bone volume (BV) and bone cortical thickness as well as the declined number of both osteoblasts and osteoclasts that resulted in low bone turnover (*Kuro-o et al., 1997*). Additionally, Klotho knockout (*Klotho*$^{-/-}$) mice established through interbreeding heterozygous *Klotho*$^{+/-}$ mice experienced severe mineralization defects (*Yuan et al., 2012*). The mineralization defects in *Klotho*$^{-/-}$ mice were possibly resulted from enhanced vitamin D signaling since Klotho could restrain trabecular bone defects due to vitamin D deficiency (*Dong et al., 2022*; *Murali et al., 2016*). However, in *kl/kl* mice, the bone phenotype did not totally resemble the age-related osteoporosis featured by both low bone volume and bone turnover (*Kawaguchi et al., 1999*; *Yamashita, Nabeshima & Noda, 2000*). The impaired bone formation and bone resorption resulted in definite low bone turnover. However, the bone volume and thickness of trabecular bone in *kl/kl* mice might increase (*Kawaguchi et al., 1999*; *Murali et al., 2016*). Long bone in *kl/kl* mice could demonstrate stretched trabeculae in the epiphyses (*Yamashita, Nabeshima & Noda, 2000*). Concurrently, the upregulated Opn expression could be accompanied by increased osteoid volume and thickness in *Klotho*$^{-/-}$ mice (*Andrukhova et al., 2017*; *Murali et al., 2016*). Yet, there is no exact explanation for the conflicting increasing or decreasing bone volume (*Yamashita, Nabeshima & Noda, 2000*). The Klotho-hypomorphic mice exhibited distinctive bone characteristics. However, *Xiao et al. (2019)* discovered that in Klotho overexpressing mice, the bone phenotype did not change obviously though some genetic variation could be detected (*Gu et al., 2019*). Using the specific knockout mouse models, *Kaludjerovic et al. (2017)* established a *Prx1Cre;Klotho*$^{fl/fl}$ mice in which Klotho was deleted in long bone mesenchyme. Yet, they did not observe variation in bone formation or BV in *Prx1Cre; Klotho*$^{fl/fl}$ mice. It was notable that under uremic condition, *Prx1Cre;Klotho*$^{fl/fl}$ mice showed significantly decreased BMD and cortical thickness accompanied by elevated osteoclasts number. More recently, *Komaba et al. (2017)* investigated the specific role of Klotho in osteocytes by generating *Dmp1Cre;Klotho*$^{fl/fl}$ mouse model. Surprisingly, they discovered increased bone formation and bone volume in the mutants, which was in contrast to the phenotype of global Klotho knockout mice. These data indicated that Klotho exerts distinct function in different cell populations. Moreover, Klotho regulates bone development and regeneration through regulating mineral homeostasis and managing bone remodeling. However, both mechanisms are not fully understood, leading to conflicting results that necessitate further investigation.

**Table 1 Summary of the relationship between Klotho level and bone health in human in different diseases.**

| Subjects | Aim | Laboratory parameters in relationship with mineral metabolism | Alternation in Klotho concentration | Results | Ref |
|---|---|---|---|---|---|
| Chronic kidney disease (CKD) patients | Evaluating BMD and trabecular bone score (TBS) in association with FGF-23 and Klotho. | **Serum Ca:** 2.35 ± 0.15 mmol/l in G1–G3; 2.43 ± 0.2 mmol/l in G4–G5. **Serum Pi:** 1.06 ± 0.02 mmol/l in G1–G3; 1.34 ± 0.05 mmol/l in G4–G5. G1 to G5 refers to progressive stages of kidney disease | 0.176 ± 0.15 ng/ml in G1–G3; 0.195 ± 0.1 ng/ml in G4–G5 | TBS between G1 and G2 exhibit significant decrease. Klotho was positively associated with TBS in the early stage of CKD while no significant connection was discovered between Klotho and BMD in those patients. | Kužmová et al. (2021) |
| One year after renal transplantation patients | Analyzing the association between bone quality and bone related molecular. | **Serum Ca:** 9.8 mg/dl (normal); **Serum Pi:** 3.1 mg/dl (normal) | 945.2 pg/ml (485.0–2044.2) | Bone densitometry (DXA) had correlations with Klotho level. Higher Klotho level in line with higher BMD. | Ferreira et al. (2021) |
| Patients who had a kidney transplant (KT) a year ago | Analyzing the correlations between Klotho and disorders in BMD. | **Serum Ca:** 9.2 ± 0.1 mg/dl; **Serum Pi:** 3 ± 0.2 mg/dl | 1.37 ± 0.13 ng/ml | Klotho is negatively related to lumbar bone density but positively associated with osteocalcin levels in KT recipients | Matei et al. (2022) |
| Middle-aged sedentary adults | Analyzing the relationship between Klotho and the BMD. | N/A | 775.3 (363.7) pg/ml in all; 814.1 (452.2) pg/ml in men; 741.4 (265.6) pg/ml in women. Datas after adjusting for lean mass index (LMI) in parentheses | BMD were significantly positively associated with plasma s-Klotho levels. However, the relationship disappeared after controlling for LMI. | Amaro-Gahete et al. (2019) |
| Northern Chinese postmenopausal women | Observing the relationship between FGF-23/Klotho and lumbar spine BMD. | **Serum Ca:** 2.28 ± 0.09 mmol/L in normal BMD postmenopausal women; 2.29 ± 0.09 mmol/L in osteopenia postmenopausal women; 2.29 ± 0.08 mmol/L in osteoporosis patients | Lg Klotho (Log-transformed Klotho): 2.34 ± 0.21 in normal BMD postmenopausal women; 2.33 ± 0.21 in osteopenia postmenopausal women; 2.28 ± 0.23 in osteoporosis patients. Significant difference ($P < 0.05$) when compared with the normal BMD group and Osteopenia group, respectively. | The Klotho protein level decreased gradually from normal bone mineral density to osteoporosis patients. Lg Klotho was positively related to LBMD in northern Chinese postmenopausal women. However, after adjusting for many potential confounders, the significant relation disappeared. | Han et al. (2019) |

(Continued)

| Subjects | Aim | Laboratory parameters in relationship with mineral metabolism | Alternation in Klotho concentration | Results | Ref |
|---|---|---|---|---|---|
| Postmenopausal women in US | Analyzing the connection between osteoporosis(OP) and Klotho. | **Serum Ca:** 2.4 ±0.1 mmol/L in non-OP; 2.4 ± 0.1 mmol/L in OP; **Serum Pi:** 1.26 ± 0.17 mmol/L in non-OP; 1.30 ± 0.18mmol/L in OP | $Log_2$Klotho pg/ml (Logarithm of plasma Klotho): 9.70 [9.40, 10.00] in non-OP; 9.57 [9.27, 9.80] in OP | 824.09pg/ml was a critical value of Klotho. When Klotho level reached the concentration, the risk of Osteoporosis decreased significantly. | *Jiang et al. (2023)* |
| Egyptian Sickle Cell Disease (SCD) Patients | Determining the potentiality for Klotho to indicate low BMD. | **Serum Ca:** 9.46 ± 0.32 mg/dL in control groups; 8.47 ± 0.44 mg/dL in SCD patients | 7.46 ± 8.81 ng/mL in control groups; 1.34 ± 1.07 ng/mL in SCD patients | SCD patients with low BMD exhibited obviously lower Klotho level. The sensitivity for Klotho to detect BMD was 94.9%, the specificity was 95.2%, and the cutoff value was 1.65 ng/mL of Klotho. | *Hamdy et al. (2022)* |
| Patients with cirrhosis | Investigating the correlations between Klotho and BMD. | N/A | 1,051 pg/ml in cirrhosis groups; 1,842 pg/ml in the control groups | Cirrhosis usually associated with higher risk of osteoporosis. In patients with cirrhosis, both BMD and the level of serum Klotho was significantly lower compared to the control group regardless of the etiology and severity. | *Katsaounis et al. (2023)* |
| People expose to work-related harmful factors of periodontitis | Analyzing the Klotho level in oral fluid and its connection with the severity of periodontitis including related bone defects. | N/A | Patients in hazardous work-related conditions exhibited lower Klotho level in oral fluid in comparison to those without this harmful factor. | Klotho might be the marker of progression in the early stage of periodontitis. It is negatively related to the severity of periodontitis. The decrease of Klotho level was in line with the increase of bone destruction marker MMP-8 and decrease of bone remodeling marker osteocalcin. | *Vozna et al. (2021)* |
| Patients with active acromegaly (Acro) | Analyzing the role of FGF-23/Klotho system on bone. | **Serum Ca:** 9.7 ± 0.08 mg/dL (normal); **Serum Pi:** 4.05 ± 0.1mg/dL (normal) | 9.15 ± 1.74 ng/mL in Acro groups | Klotho were related to osteocalcin positively. Klotho did not differ significantly in Acro groups with or without abnormal bone mass. Klotho presented no connection with GH/IGF1 axis and had limited role in the follow-up of acromegalic patients. | *Bilha et al. (2021)* |
| A 13-month old girl with a balanced translocation between chromosomes 9 and 13: t(9,13)(q21.13; q13.1) | Describing a new disease caused by increasing sKL level resulted from mutation. | **Serum Ca:** 9.4 mg/dl (normal: 9.0–10.5); **Serum Pi:** 2.1 mg/dl (normal: 4–7 in child) | Used Western Blot to detect the amount of Klotho in plasma of patient and found that it increased significantly. | Klotho might play a role in renal osteodystrophy. | *Brownstein et al. (2008)* |

Additionally, recent research discovered the anti-aging role of Klotho in long bones. *Roig-Soriano et al. (2023)* established the SAMP8 mice model that exhibits aging characteristics. They transfected Klotho to aging mice through adeno-associated virus (AAV) vectors to enhance Klotho expression. The cortical expansion caused by aging was reversed, which proved that Klotho could alleviate the age-related bone changes.

Moreover, serum Klotho level could be influenced by diseases and lead to damaged bone. We summarized studies that examined the relationship between Klotho level and human bone health under different diseases in Table 1. However, it should be noted that due to the improvable performance and reproducibility in current commercially available assays and different processing methods for the collected serum, human serum Klotho level varies widely among different studies (*Neyra, Hu & Moe, 2021*).

## Craniofacial bone

### Craniofacial bone alternations in Klotho-deficient mice

To investigate the specific role of Klotho in craniofacial region, we established the $Osx^+$-progenitor targeted Klotho ablation mouse model using the Cre-LoxP recombination system (*Fan et al., 2022*). The morphological changes of the jawbone in *OsxCre;Klotho$^{fl/fl}$* mice were consistent with that of the long bone reported by *Kuro-o et al. (1997)*. In *Osxcre; Klotho$^{fl/fl}$* mice, the formation and activity of osteoblasts were negatively affected and osteoclastogenesis was inhibited, indicating low bone turnover phenotype. Since the inhibitory effect on osteoclasts was significantly more obvious than the reduction in osteogenesis, the alveolar bone volume in the mutants increased (*Fan et al., 2022*). Moreover, the senescence-accelerated mouse prone 1 (SAMP1) model is established to mimic mice aging. In *SAMP1/Klotho$^{-/-}$* mice, the mandibular ramus displayed dysplastic bone development, calcified skeletal structures and a widespread ground glass appearance (*Yook et al., 2020*). Furthermore, *Hikone et al. (2017)* summarized the morphological alternations of alveolar bone in *kl/kl$^{norpi}$*, *αKlotho$^{-/-}$* and *kl/kl$^{lowpi}$* mice. Both *kl/kl$^{norpi}$* and *αKlotho$^{-/-}$* mice exhibited intensive hematoxylin-stained areas with amorphous materials indicating histological changes. Besides, the osteoclasts and osteoblasts were distributed dispersedly with Dmp1 gathered at several empty osteocytic lacunae. Interestingly, elevated serum Pi was detected in Klotho hypomorphic mice. A low Pi diet could improve histological abnormalities in *kl/kl* mice. But in *αKlotho$^{-/-}$* mice, the aberration cannot be rectified by a low Pi diet. *kl/kl$^{lowPi}$* mice established an elevated α-Klotho level and α-Klotho gene expression can rescue the altered phenotype in *kl/kl* mice. As a result, besides phosphate metabolism abnormality, the Klotho/FGF23 signaling pathway may also take action on the discrepancy of the craniofacial bone.

In addition, bone and teeth possess some similar morphological features, for example, comparable extracellular matrix components as well as mineralization processes. *Suzuki et al. (2008)* have confirmed that Klotho may be involved in dentin formation and mineralization. However, further research on how Klotho affects dental hard tissue is still needed.

### Role of Klotho in craniofacial bone under inflammation

Infection from the tooth, periodontal tissues or blood could affect the craniofacial bone, resulting in local inflammation and bone defects. Periodontitis or apical periodontitis (AP) are the general causes of oral bone loss in clinical practice. Studies illustrated that severe periodontitis featured by alveolar bone destruction affects more than 7.8% of adults in the U.S. and more than 52% of adults have at least one tooth suffering from AP (*Tibúrcio-Machado et al., 2021*; *Brenchley et al., 2024*; *Zhao et al., 2024*). Thus, discussing the impact of Klotho on maxillofacial bone reconstruction under the condition of inflammation is necessary.

To simulate bone loss and regeneration under inflammatory condition, we developed mice models of AP and alveolar bone healing following tooth extraction. *OsxCre;Klotho*^fl/fl^ mice under inflammation encountered more remarkable alveolar bone loss and root shortening in comparison to WT mice. After tooth extraction, WT mice completed bone healing in 21 days while the mutant mice needed more time to recover. Markedly decreased bone volume/tissue volume (BV/TV) and BMD were discovered in the lately formed bone of mutant mice. Meanwhile, alveolar bone in the sockets of these mice was less interconnected, with increased bone marrow area. All the outcomes determined that Klotho could reduce bone loss and promote craniofacial bone repair under inflammatory condition (*Fan et al., 2022*). This was the first study that emphasized the pivotal role of Klotho in inhibiting bone resorption and facilitating bone repair in inflammatory states. Nevertheless, further studies are needed to validate the consequence. In addition, more research concerning other skeletal regions may broaden the scope of the result.

Furthermore, Klotho is tightly associated with oral diseases. Lacking serum α-Klotho may lead to severe periodontitis, raise the risk of suffering from dental caries and increase the possibility of losing teeth (*Chen et al., 2022a*). The declining oral health status may increase the incidence of dentinal and periodontal diseases and induce bone damage caused by inflammation.

Concurrently, studies have reported the bone morphology of Klotho hypomorphic mice and Klotho globally knockout mice. Moreover, Klotho-specific ablated mice including *Prx1Cre;Klotho*^fl/fl^ mice, *Dmp1Cre;Klotho*^fl/fl^ mice and *OsxCre;Klotho*^fl/fl^ mice that target different cells and bones were generated. We summarized the bone phenotypes in different mouse models in Table 2.

## Function of Klotho on diverse cells in bone development and regeneration

Klotho is expressed in mesenchymal stem cells (MSCs), osteoblasts, osteoclasts and osteocytes, which play essential roles in bone development and regeneration. It could also direct stem cell fate to regulate bone development and repair. In the craniofacial region, Klotho has been identified in the stem cell populations in bone marrow and periodontal ligament (PDL). Here, we summarized the recent advances on the role of Klotho in diverse cells during bone development and repair.

**Table 2 Different bone phenotypes in Klotho target ablation mice.**

| Genotype | Bone formation | Bone resorption | Mineralization | Homeostasis | Bone phenotype | References |
|---|---|---|---|---|---|---|
| **Long bone** | | | | | | |
| kl/kl | Inhibited decreased osteoblasts number | Inhibited decreased osteoclasts number | Hypomineralization | Increased serum Pi and slightly higher Ca | Bone volume ↓ BMD ↓ Low bone turnover cortical thickness ↓ | *Kuro-o et al. (1997)*, *Minamizaki et al. (2018)* |
| Klotho⁻/⁻ | Inhibited decreased bone formation rate | Inhibited decreased osteoclasts number | Hypomineralization | Increased serum Pi and Ca | Bone volume ↓ BMD ↓ Cortical thickness ↓ | *Yuan et al. (2012)*, *Andrukhova et al. (2017)* |
| βactinCre; Klothoᶠˡ/ᶠˡ | Inhibited decreased osteoblast marker genes | N/A | Hypomineralization | Increased serum Pi and Ca | N/A | *Li et al. (2013a)* |
| Prx1Cre; Klothoᶠˡ/ᶠˡ | No significant differences in bone formation rate | Activated increased osteoclasts number | No significant alternation | Normal serum Pi and Ca | In uremic mice: BMD ↓ Cortical thickness ↓ | *Kaludjerovic et al. (2017)* |
| Dmp1Cre; Klothoᶠˡ/ᶠˡ | Activated increased bone formation rate | No significant differences | Hypermineralization | Normal serum Pi and Ca | Bone volume ↑ | *Komaba et al. (2017)* |
| **Craniofacial bone** | | | | | | |
| OsxCre; Klothoᶠˡ/ᶠˡ | Inhibited decreased osteoblasts number | Inhibited decreased osteoclasts number | No significant alternation | Normal serum Pi and Ca | Bone volume ↑ BMD in regenerative bone after tooth extraction ↓ Low bone turnover | *Fan et al. (2022)* |

## Bone marrow stem cells

MSCs are multilineage cells that could self-renew and differentiate into various cell types which are important for tissue regeneration (*Bacakova et al., 2018*). Low vitality in stem cells is one of the typical symptoms of aging (*López-Otín et al., 2013*). Once injured, stem cells migrate toward damaged tissue and differentiate into target cells (*Fu et al., 2019*). Bone marrow stem cells (BMSCs) are the most commonly used stem cells in tissue repair (*Fu et al., 2019*). *Kaludjerovic et al. (2017)* conducted 5/6 nephrectomy surgery on *Prx1Cre; Klothoᶠˡ/ᶠˡ* mice to establish a uremic mice model, in which Klotho in long bone BMSCs was specifically knocked out. They found a higher osteoclast number and higher osteoclast surface which is on behalf of the high degree of osteoclast differentiation in the mutant group. Receptor activator of nuclear factor-κB ligand (Rankl) and Opg are key factors that mediate bone resorption secreted by osteoblasts. The Rankl/Opg ratio turned out to be higher in *Prx1Cre;Klothoᶠˡ/ᶠˡ* mice femur, strengthened that Klotho in Prx1⁺ BMSCs inhibited osteoclast differentiation in mice with renal insufficiency. *βactinCre;Klothoᶠˡ/ᶠˡ* mice is a global Klotho deletion mice that crosses *Klothoᶠˡ/ᶠˡ* mice with mice that expresses Cre recombinase under the β-actin promotor of human. In *βactinCre;Klothoᶠˡ/ᶠˡ* mice, the BMSCs tended to differentiate into adipocytic colonies compared to osteoblastic colonies (*Li et al., 2013b*). Consisting with this, the osteoblastic marker genes like *Runx2, Opn, Alp*

and *Ocn* were lower in mutant mice while the adipocyte marker gene *Pparγ* was higher. The osteogenesis or adipogenesis of BMSCs directly affect bone morphology (*Li et al., 2011*). As a consequence, both the formation and resorption of bone were restrained in *βactinCre;Klotho^{fl/fl}* mice with the accumulation of bone marrow fat. *Feng et al. (2023)*, confirmed this outcome and illustrated that BMSCs were hyperactive in *kl/kl* mice, yet their functions were weakened. The potential for adipogenesis was strengthened while the osteogenic genes *Alp* and *Ocn* decreased remarkably. *In vivo* experiments demonstrated that BMSCs in the appendicular skeleton of *kl/kl* mice were probably in an active cell cycle and twice more than the WT mice. Isolated BMSCs from *kl/kl* mice revealed a tendency towards hyperproliferation and exhaustion. Adding sKL into the culture medium reduced the hyperproliferation of BMSCs in *kl/kl* mice, which might be associated with mTORC1 signaling (*Feng et al., 2023*). However, *in vitro* experiments suggested that using recombinant Klotho (rKL) promoted BMSCs proliferation but reduced mineralization and inhibited the expression of genes associated with osteoblasts. The results suggested that *Runx2*, *Alp*, *Ocn* and *Col1a1* were apparently lower in cells co-cultured with rKL than the control.

### Osteoblasts

By modulating bone modeling and remodeling, osteoblasts are indispensable in maintaining bone homeostasis (*Ponzetti & Rucci, 2021*). It is still controversial whether Klotho plays a positive or negative role during osteogenic differentiation. Diverse investigations came up with opposite results. Some reports considered Klotho as a inducing factor in osteoblastic differentiation. *Toan et al. (2020)*, transfected sKL into MC3T3-E1 cells to investigate how overexpressing Klotho affects osteogenic differentiation. After 3 days of osteogenic induction, MC3T3-E1 with overexpressed Klotho showed an increase in osteoblast markers (*Bmp2*, *Runx2*, *Alp*, *Opn*, *Col1a* and *Ocn*). The mineralization was evidently enhanced as well. This was in accord with the *in vitro* outcome of osteoblasts from the calvaria bone. Conversely, some reports indicated Klotho plays as an inhibiting factor in osteoblastic differentiation. *Komaba et al. (2017)* applied the lentiviral vector system to induce Klotho overexpression in ME3T3-E1 cells. The alizarin red staining (ARS) results declared that the Klotho overexpression group revealed delayed osteogenic. Also, *Murali et al. (2016)* discovered no significant alternations in ALP activity in osteoblasts isolated from *Klotho^{−/−}* mice. Indeed, the culture condition, cell activity and the duration and concentration of Klotho may influence the osteoblasts' function.

### Osteocytes

Osteocytes are the most profuse cells in bone that can deliver signals of skeleton formation and resorption or leave the bone lacuna and convert into osteoblasts to participate in bone remodeling (*Bonewald, 2008*). In *Dmp1Cre;Klotho^{fl/fl}* mice, the bone formation and bone mass were increased. Furthermore, overexpressing Klotho suppressed mineral deposition and osteogenic activity in osteoblast cell lines during osteocyte differentiation (*Komaba et al., 2017*). The osteocyte-specific Klotho deletion promoted bone formation while not

affecting bone resorption (*Komaba & Lanske, 2018*). This result suggests that Klotho might have specific functions in diverse cells in long bone.

### Osteoclasts

Osteoclasts derive from monocyte/macrophage hematopoietic precursors and take a decisive action on bone resorption (*Boyle, Simonet & Lacey, 2003*). Klotho was detected in osteoclasts, suggesting a direct regulatory action of Klotho on osteoclastic differentiation (*Raimann et al., 2013*). The number of osteoclasts decreased significantly in $Klotho^{-/-}$ mice (*Andrukhova et al., 2017*). Interaction between osteoclasts and osteoblasts regulate the remodeling of bone (*Kim et al., 2020*). *Kawaguchi et al. (1999)* demonstrated that the decline of bone resorption in *kl/kl* mice resulted from the anomalous osteoclast differentiation from the progenitor rather than the flaws in the existing osteoclasts. Consistent with the outcome, using osteoclast differentiation factor Rankl to induce osteoclastogenesis in bone marrow macrophages (BMMs) of both Klotho-hypomorphic mice and WT mice. *Yu et al. (2021)* discovered that the osteoclasts number in the *kl/kl* mice diminished obviously. The consequence suggested that Klotho promotes osteoclastogenesis stimulated by Rankl. In $Klotho^{-/-}$ mice, not only osteoclastogenesis but also B lymphopoiesis was suppressed. B-lymphoid lineage cells may manage osteoclastogenesis. These cells could either promote osteoclasts differentiation through expressing ODF/RANKL or act as the osteoclast progenitors. The expression of ODF/RANKL and B lymphopoiesis were inhibited in $Klotho^{-/-}$ mice, so osteoclastogenesis was negatively influenced (*Manabe et al., 2001*).

### Adipose-derived stem cells

Adipose-derived stem cells (ADSCs) are abundant and more acquirable *via* liposuction (*Doi et al., 2013*). Klotho promoted adipogenic differentiation of ADSCs (*Fan & Sun, 2016*). In a mouse model that overexpressed mKL and sKL, *Ocn* and *Opn* declined while the adipocyte maker gene *PPARγ, aP2* and *Lpl* increased markedly, indicating an enhanced adipogenesis in bone marrow. However, compared with WT mice, the bone mass, BMD as well as the cortical bone thickness have no significant differences in $Klotho^{Tg}$ mice in which the *hEF1a* regulates the overexpression of mKL (*Xiao et al., 2019*). $Klotho^{-/-}$ ADSCs exhibited declined telomerase liveliness and length, which diminished the activity of ADSCs (*Ullah, Sun & Hare, 2019*).

### Orofacial bone marrow mesenchymal stem cells

We targeted ablated Klotho in Osx⁺-mesenchymal progenitor and found Klotho deletion in orofacial bone marrow mesenchymal stem cells (OMSCs) diminished Osx immunoreactivity significantly while Runx2-positive cells were unchanged. Generally, Runx2 represents the progenitor cells differentiated to preosteoblasts, while Osx expression indicates the trends to osteoblasts. The results illustrated that Klotho ablation principally affects the maturation and activity of osteoblasts (*Fan et al., 2022*). Incubating calvaria osteoblasts suggested that the osteoblast mineralization in $Klotho^{-/-}$ mice decreased markedly (*Yuan et al., 2012*). Since we found that target ablation of Klotho in Osx⁺-mesenchymal progenitor influences osteoblast activity and maturation, we further
transfected lentiviral to primary calvarial osteoblasts from *OsxCre;Klotho*<sup>fl/fl</sup> mice to overexpress Klotho and osteogenic inducted them. Markedly increased ALP and ARS intensities were detected in overexpression Klotho osteoblasts, together with higher expression levels of osteogenesis-related markers. This reinforced that Klotho plays a role in promoting MSCs differentiation towards osteoblastic lineage cells (*Fan et al., 2022*).

In *OsxCre;Klotho*<sup>fl/fl</sup> mice, the number of TRAP[+] osteoclasts/bone surface diminished notably in the orofacial bone. Osteoclast differentiation and activation marker gene such as *Mmp9* and *Acp5* were inhibited in mutant mice. We determined the decreased Rankl and increased Opg, leading to lower Rankl/Opg ratio. Through the co-cultivation of osteoblasts generated from calvaria and bone-marrow-derived osteoclasts, we discovered that Rankl was upregulated in osteoblasts that overexpressed Klotho however downregulated in Klotho-ablated cell. Transfecting Klotho in *Klotho*<sup>fl/fl</sup> mice osteoblasts led to increased Rankl. These results revealed that suppressed osteoclastogenesis was associated with Rankl/Opg expression in OMSCs regulated by Klotho.

Under inflammatory conditions, the reduced osteogenic potential of OMSCs was detected in mutant mice. Furthermore, in the extraction sockets, Runx2[+]-osteoblasts decreased remarkably in Klotho-deficient mice compared with WT group, indicating Klotho ablation inhibited osteoblast differentiation of OMSCs during repair. However, in contrast, the activity of osteoclasts increased markedly in mutant mice. By adding TNF-α in the osteoblasts-osteoclast co-culture system to imitate the inflammatory microenvironment, we found that TNF-α strengthened osteoblast-induced osteoclastogenesis. This trend was even more notable in mutant mice, indicating that the mandibular alveolar bone was repaired by Klotho through directly promoting osteogenic differentiation as well as suppressing osteoblasts Rankl expression to reduce osteoclastogenesis (*Fan et al., 2022*).

### Human periodontal ligament stem cells

Human periodontal ligament stem cells (hPDLSCs) are dental tissue derived MSCs. They are more accessible with higher cell growth possessed in comparison to BMSCs. hPDLSCs have multilineage differentiative potential and extraordinary therapeutic abilities in orofacial diseases. Periodontal ligament stem cells (PDLSCs) are the main candidate stem cells to treat bone loss caused by periodontitis (*Eleuterio et al., 2013*; *Ou et al., 2019*; *Trubiani et al., 2019*). Klotho protected PDLSCs from oxidative stress to promote osteogenesis. *Chen et al. (2019b)* first identified that rKL might protect hPDLSCs from $H_2O_2$-induced oxidative stress. During $H_2O_2$-induced oxidative stress, Klotho could regulate the function of mitochondria and monitor the antioxidant system through the PI3K/AKT/FoxO1 pathway. Furthermore, *Zhu et al. (2021)* pointed out that when exposed to Klotho, the $H_2O_2$-induced oxidative stress and apoptosis were inhibited by si-UCP2 in hPDLSCs. This indicates that recombinant human Klotho prevents $H_2O_2$-induced oxidative stress and apoptosis in hPDLSCs by upregulating UCP2. Recently, *Niu et al. (2023)* confirmed that Klotho arranged the immune-regulatory effect of hPDLSCs through managing macrophages, which stimulated anti-inflammatory M2 polarization and subdued proinflammatory M1 polarization. Meanwhile, Klotho inhibited the hyperactivity

of autophagy in hPDLSCs. Thus, Klotho could improve the regenerative function of hPDLSCs.

## Mechanism

Klotho is involved in various signaling related to bone modeling and remodeling (*Abolghasemi et al., 2019*; *Luo et al., 2022*). A lot of research focused on the mechanism of the bone-regulating role of Klotho. Currently, three signaling pathways including FGF-23, NF-κB and Wnt are the most widely studied in bone development and regeneration. Moreover, the mechanisms of Klotho in regulating craniofacial bone focus on certain pathological states such as inflammation and oxidation. Thus, TNFR and PI3K/AKT are the top two widely studied signaling pathways in explaining how Klotho directs craniofacial bone formation and regeneration. These signaling pathways are activated under different conditions to regulate bone formation, resorption and mineralization, thereby altering the skeletal phenotype or promoting bone repair. We summarized the regulatory signaling pathways and concluded their role in long bone and maxillofacial bone here.

### FGF-23

The major role of α-Klotho is binding to FGFR1 to enhance the affinity between FGF-23 and FGFR1 (*Chen et al., 2018*; *Erben, 2017*). FGF-23 gene mutation would cause serious bone diseases (*Tresguerres et al., 2020*). As a multifunctional growth factor, FGF-23 is produced by osteocytes and osteoblasts and is considered as a bone-derived hormone. It is a key factor in bone modeling and remodeling by regulating mineral homeostasis. Mineral metabolic disorders could lead to abnormalities of bone morphology (*Clinkenbeard, 2023*; *Lu & Feng, 2011*). To reduce serum phosphate and $1,25(OH)_2D$ levels, FGF-23 targets FGFR1 at the proximal of the kidney (*Han & Quarles, 2016*; *Shimada et al., 2003*). In α-Klotho deficient mice, their serum vitamin D and phosphate increased remarkably consistent with the phenotype of $Fgf23^{-/-}$ mice (*Nakatani et al., 2009*). In Klotho overexpressing mice, the FGF-23 level rose significantly, accompanied by hypophosphatemia (*Brownstein et al., 2008*). In fact, the FGF-23/FGFR/a-Klotho ternary complex in human was considered to coordinate bone mineral metabolism and phosphate metabolism in the kidney (*Quarles, 2019*). Klotho-FGFR1 complex could control FGF-23 production and secretion by establishing an autocrine feedback circle. *In vitro* experiments suggested that only Klotho-transfected MC3T3-E1 cells treated by FGF23 exhibited increased FGF23 mRNA expression while no significantly higher FGF23 mRNA expression can be detected in GFP-transfected cells. Further *in vivo* research illustrated that during renal failure, Klotho in bone was important in inducing FGF-23 production since long bone-specific Klotho ablation mice failed to increase FGF-23 level (*Kaludjerovic et al., 2017*).

Two downstream targets of the FGF-23 signaling were studied, including EGR and ERK. EGR is one of the downstream markers of the FGF-23 signaling pathway. Klotho deficiency reduced the expression of EGR-1 (*Shalhoub et al., 2011*). Moreover, sKL could

directly stimulate the EGR signaling pathway. *Toan et al. (2020)* used sKL and siEGR-1 to incubate MC3T3-E1 cells. Three days after culture, the sKL group had the highest trend in osteogenic differentiation followed by the sKL+siEGR-1 group. siEGR-1 group had the worst osteogenic differentiation trend. The outcome made it clear that sKL is related to the regulation of EGR. To further investigate the role of EGR, they knocked out the gene and discovered that EGR-1 assisted osteoblast differentiation. The result is in parallel with the role of Klotho in promoting bone formation. Besides, they first discovered the key regulatory site: pGL3-EGR-1 SacI promoter region (a 130 bp region range between 415 and 282 bp), further explaining EGR as a downstream target of Klotho in bone remodeling.

ERK is another mediator in the FGF-23 signaling. It is significant for osteoblast and osteocyte differentiation and mineralization (*Kono et al., 2007*; *Kyono et al., 2012*). Conducting fibroblast growth factor-basic (bFGF) treatment on long bone-specific deletion of Klotho mice that had mature osteocytes revealed that bFGF treatment elevated FGF23 mRNA expression as well as the phosphorylation of downstream ERK (*Kaludjerovic et al., 2017*). *Shalhoub et al. (2011)* suggested that *in vitro* incubation of MC3T3-E1 cells with the combination of FGF-23 and sKL activated ERK signaling to promote Klotho-induced MC3T3-E1 cell proliferation yet inhibited MC3T3-E1 mineralization. Moreover, *Zhang et al. (2015)* esteemed a consistent result by incubating BMSCs with sKL. According to their *in vitro* experiments, sKL supported the proliferation of BMSCs and suppressed its osteogenic differentiation. However, this effect was achieved through inhibiting FGFR1/ERK signaling. Whether ERK promotes or inhibits bone formation is uncertain (*Kono et al., 2007*; *Ge et al., 2007*). The discrepancy might be due to different effects of ERK signaling at different stages of osteogenesis. *Minamizaki et al. (2018)* discovered that sKL treatment in *kl/kl* mice caused a mineralization defect. They further detected defective matrix mineralization in rat calvarial (RC) cells with sKL plus FGF23. By detecting ERK phosphorylation, the result turned out that sKL activates ERK pathway in *kl/kl* RC cell osteogenic model with FGF-23. Notably, although FGF23 production kept high in *kl/kl* until at least 48h in calvaria organ cultures, the increased expression of pERK1/2 was detected only on treatment with sKL. These results suggested that sKL directly stimulates FGF23 signaling through ERK in bones. Phex is connected to bone mineralization. Its haploinsufficiency causes human osteomalacia, a hypomineralization disease (*Guo & Quarles, 1997*; *Miao et al., 2001*). Moreover, RT-qPCR confirmed Phex as a downstream effector that is downregulated during activated p-ERK1/2 in the signaling pathway sKL-FGF23-FGFR of osteoblasts (*Minamizaki et al., 2018*). Notably, these outcomes conflicted with the promotion role of Klotho for osteoblasts. On one side, the role of Klotho in osteoblast mineralization is controversial currently. On the other side, *in vitro* experiments could not completely represent the physiological processes *in vivo* and Klotho hypomorphic mice may not exhibit normal response to sKL.

### NF-κB

NF-κB signaling pathway controls osteoclastogenesis to regulate normal bone modeling and remodeling in the impaired skeleton (*Boyce & Xing, 2008*). It is also an important

signaling pathway in inflammation (*Wu et al., 2023*). Rankl can connect to the extracellular part of Rank and stimulate NF-κB signaling pathway to promote osteoclastogenesis (*Xing, Schwarz & Boyce, 2005*). *Yu et al. (2021)* selected mice femur BMMs from *kl/kl* mice and WT mice to conduct *in vitro* experiments. The outcome suggested that NF-κB signaling pathway was the most enriched pathway. The consequence exhibited that a deficiency of Klotho impairs osteoclastogenesis. Exogenous Klotho could improve osteoclastogenesis. The maintenance of the resorption function of osteoclasts is important for bone homeostasis. After analyzing the RANK protein structure and extracellular region of Klotho and using a specific program to explore the possible combined forms, they established a RANK/Klotho complex. Immunoprecipitation (Co-IP) confirmed the existence of the complex. However, it is notably that the downstream molecular of NF-κB signaling IκB and p65 was not significantly altered in BMMs. In *Klotho^{Tg}* mice, only the Klotho isoform contained KL1 obviously decreased the NF-κB activation mediated by both basal and TNF-α or FGF23. These findings recommended that the KL1 domain contained suppressive functions on both TNF-α and FGF23 (*Xiao et al., 2019*). In *Dmp1Cre;Klotho^{fl/fl}* mice, no detectable Rankl/Opg alternation was found. These results illustrated the various functions of Klotho in the altered cell types (*Komaba & Lanske, 2018*; *Komaba et al., 2017*).

Klotho could also prevent the apoptosis of MC3T3-E1 cells through regulating NF-κB signaling pathway. Dexamethasone (DEX) pretreatment to MC3T3-E1 cells stimulated the translocation of NF-κB and the degradation of its inhibitor IκBα and induced decreased caspase3 and increased cleaved-caspase3. Klotho transfection could inhibit the above apoptosis-related processes induced by DEX. Upon the activation of NF-κB, IκBα phosphorylated and degraded to IκBα- NF-κB complex and released p65 to translocate into the nucleus. Western Blot and immunofluorescence results suggested that DEX induced a diminished IκBα, a raised P-p65 level and an increased nuclear translocation. Klotho partially restored the degraded IκBα and inhibited P-p65 into the nucleus. Moreover, NF-kB inhibitor pyrrolidine dithiocarbamate (PDTC) pretreated DEX-induced osteoblasts exhibit a 50.7% decline in cleaved-caspase-3 protein in comparison to DEX-only groups. In DEX-induced osteoblasts treated with Klotho, cleaved-caspase-3 protein exhibited a 73.3% down-regulation compared to the control groups. All the outcomes illustrated that the inhibition role of Klotho in ME3T3-E1 cells apoptosis at least partly depends on the NF-κB signaling pathway (*Liang et al., 2018*).

## Wnt

Wnt signaling pathway is pivotal in both the modeling and remodeling of bone. In bone regeneration, Wnt is involved in regulating both intramembranous and endochondral ossification (*Marini et al., 2023*). Klotho could target the Wnt signaling pathway, which has been proven and widely studied in several tissues excluding bone (*Tang et al., 2016*; *Zhu et al., 2021*). *Liu et al. (2007)* discovered that in *kl/kl* mice, Wnt signaling pathway was strengthened in bone. Dickkopf-1 (Dkk1) is an inhibitor of Wnt. *Dmp1Cre;Klotho^{fl/fl}* mice demonstrated a decreased expression of Dkk-1, indicating promoted osteoblastic activity in osteocyte-specific Klotho deficient mice (*Komaba et al., 2017*). Exposed UMR106-01 cells to FGF-23 and sKL in combination or alone and detected the mRNA levels of

p-β-catenin/β-catenin ratio and Dkk1 protein level after 48 h. The result proposed that no obvious discrepancy could be detected in mRNA level by exposing UMR106-01 cells to FGF-23 or sKL respectively. However, if UMR106-01 cells were cultured with FGF-23 and sKL together, increased p-β-catenin/β-catenin and Dkk1 could be recognized (*Carrillo-López et al., 2016*). In osteoblasts, FGF23/sKlotho could activate ERK phosphorylation to promote DKK1 secretion. DKK1 can inhibit β-catenin through paracrine and autocrine pathways to suppress Wnt signaling pathway (*Carrillo-López et al., 2016*). All the outcomes indicated that Klotho deficiency enhances the role of FGF-23 in enriching Wnt signaling and Klotho may regulate bone formation through Wnt signaling pathway.

Figure 2 is a summary of the cellular mechanism of Klotho in regulating long bone modeling and remodeling. However, those signaling pathways usually solely consider the direct effect of molecules on stem cells or bone cells that results in alteration in bone phenotype and ignore the impact of the changes in microenvironment under pathological conditions. Yet, this gap was subsequently described within the following exploration concerning maxillofacial bone development and regeneration.

### TNFR

Inflammatory factors can directly act on bone cells to regulate cell fate. TNF-α belongs to TNF/TNFR cytokine superfamily known as a pro-inflammatory cytokine, which could contribute to inflammatory diseases in bone and manage bone modeling and remodeling (*Balkwill, 2006*; *Rani et al., 2010*; *Tsai et al., 2014*). By binding to TNFRI or TNFRII, TNF-α could either influence the immune system or affect the skeleton system. By binding to TNFRI, TNF-α could enhance Rankl-induced osteoclastogenesis while TNFRII mainly impacts the immune system (*Zhang et al., 2001*; *Wang et al., 2020*). In fact, TNF-α promotes osteoclastogenesis through two methods-with or without Rankl to stimulate bone resorption (*Rani et al., 2010*; *Yao, Getting & Locke, 2021*). The ability of Rankl to stimulate osteoclastogenesis is far stronger than the independent role of TNF-α to directly stimulate osteoclastogenesis through increasing the precursors of osteoclasts. Rankl dependent TNF-α promotes osteoblasts, stromal cells *etc.*, to secret Rankl and finally supports osteoclastogenesis (*Yao, Getting & Locke, 2021*; *Zhao, 2020*). In bone defects caused by chronic inflammation, TNF-α inhibits osteogenesis and negatively regulates bone homeostasis (*Zhao, 2017*). Klotho is an anti-inflammatory factor whose role in reducing inflammation has been reported (*Krick et al., 2018*; *Junho et al., 2022*). In mice with AP, the bone suffered a more severe loss in *OsxCre;Klotho^{fl/fl}* mice (*Fan et al., 2022*). In our *OsxCre;Klotho^{fl/fl}* and *OsxCre* mice model with AP, the serum Ca, Pi and FGF-23 levels were normal, making it possible to detect the isolated tissue-specific function of Klotho. We observed that Klotho could regulate bone repair under the inflammatory microenvironment through the inflammation signaling pathway. Despite the bone phenotype above that confirming the result, we overexpress Klotho in the cytomembrane of osteoblasts by administrating adenovirus-mediated Cre (Ad-CRE) and set up an osteoblast-osteoclast co-culture system, then added TNF-α into the co-culture system. The outcome spotted a higher TNFRI expression in osteoblasts cytomembrane treated with TNF-α. In parallel with the result, overexpressing Klotho impaired the trend. Identical to

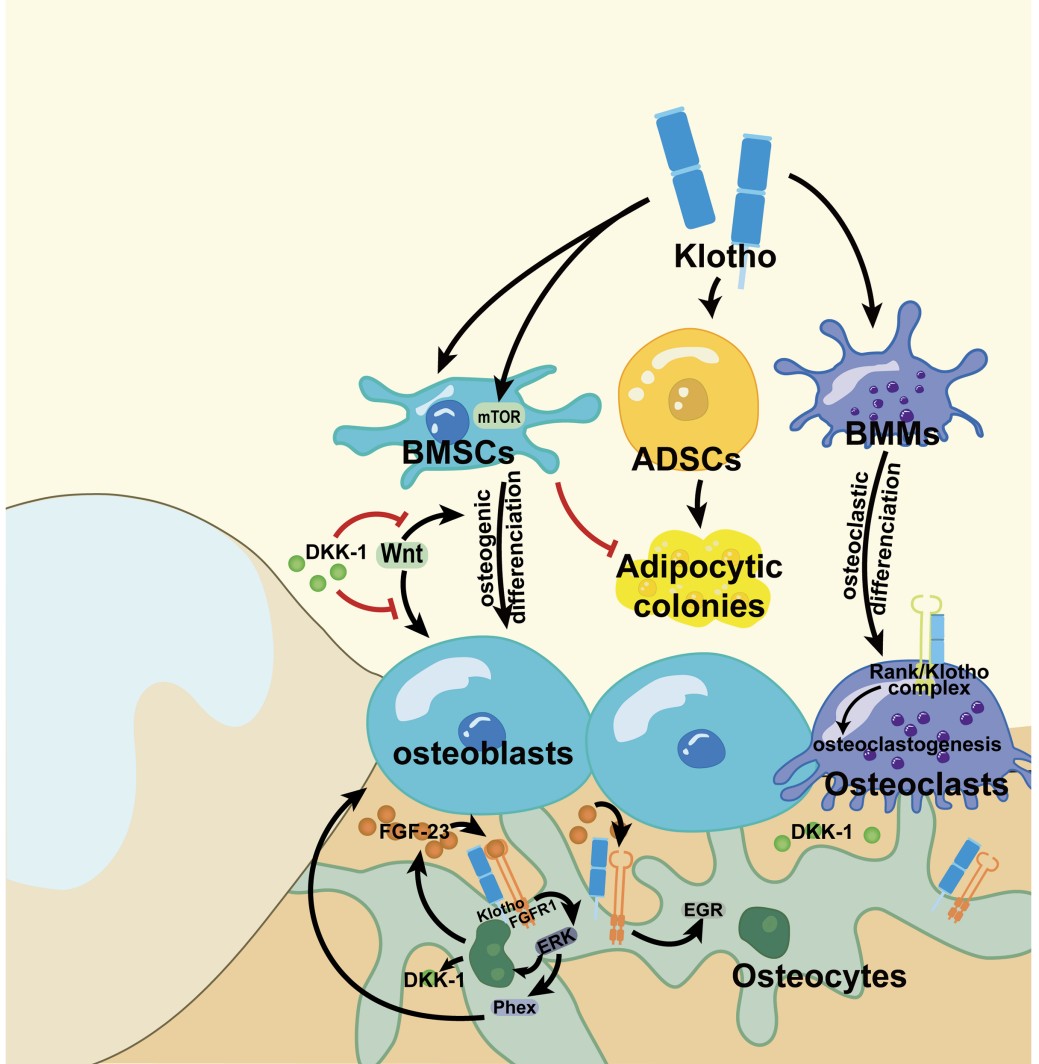

**Figure 2 Mechanism of Klotho in regulating diverse cells in long bone.** Klotho could promote the osteogenic differentiation of BMSCs and inhibited its adipocytic colonies. Klotho activated mTOR to weaken the autophagy in BMSCs as well. Yet it increases the adipocytic colonies from the ADSCs. Meanwhile, Klotho enhances the osteoclastic differentiation potential of BMMs. However, by binding to FGFR1, Klotho elevated the affinity between FGF-23 and FGFR1 to reinforce the secretion of ERK and EGR. ERK induced the expression of DKK1, which inhibited the Wnt signaling to restrain the osteogenic differentiation of BMSCs. Moreover, ERK negatively regulates its downstream effector Phex and leads to attenuated mineralization in bone. EGR is a positive regulator for osteoblasts differentiation. Additionally, Klotho can form a complex with RANK and promote bone resorption lead by osteoclasts. BMSCs, bone marrow stem cells; ADSCs, adipose-derived stem cells; BMMs, bone marrow macrophages. Figure created using Adobe Illustrator.           

this, by co-culturing Klotho-ablated osteoblastic cells that were pretreated with TNF-α with osteoclast, we discovered that osteoclast number raised a lot accompanied with the upgrades of Rankl. Coimmunoprecipitation confirmed that Klotho could bind to TNFRI directly, the combination may interfere TNFRI signaling pathway. R-7050 prevents the interaction between TNFRI and intercellular adapter molecular (*King, Alleyne & Dhandapani, 2013*). Without the Klotho ablation, the administration of R-7050 inhibited

Rankl expression induced by TNF-α, illustrating that TNFRI itself was immediately impacted by Klotho (*Fan et al., 2022*). By directly binding Klotho and TNFRI in osteoblasts, the NF-κB nuclear translocation induced by TNF-α administration was suppressed. Then Rankl expression was also restrained. The result defined that Klotho could act through TNF-α signaling induced NF-κB p65 subunit nuclear translocalization to decrease the portion of activated osteoblasts or it can directly inhibit Rankl signaling to inhibit osteoclastogenesis and accelerate bone repair (*Fan et al., 2022*).

### PI3K/AKT

hPDLSCs are commonly employed in bone tissue engineering to encourage the regeneration of new bone. However, once they are isolated from the original tissues, reactive oxygen species will be produced due to improper conditions *in vitro*. Then DNA damage is caused and apoptotic pathways are activated (*Choo et al., 2014*; *Danisovic et al., 2017*). Through $H_2O_2$-induced oxidative stress in cells, *Chen et al. (2019a)* found that Klotho reduced the oxidative stress and restored the capability to osteogenic differentiation in hPDLSCs. The activation of PI3K/AKT signaling is associated with osteogenic differentiation and osteoclastic differentiation (*Fang et al., 2022*). Additionally, PI3K/AKT is closely related to the proliferation and differentiation of stem cells. Its downstream molecular FoxO1 is closely related to oxidative stress. Western Blot analysis demonstrated that $H_2O_2$ can stimulate the AKT/FoxO1 pathway in hPDLSCs, and reduce the expression of FoxO1, resulting in decreased expression of Catalase and MnSOD that suppress superoxide. However, rKL pretreatment in hPDLSCs can reverse this trend and enhance FoxO1-mediated antioxidant expression. High expression of AKT/FoxO1 signaling pathway can also be detected in chronic AP. Its expression is higher in periapical cysts than in periapical granulomas. AKT is activated and FoxO1 is phosphorylated in periapical cysts. hPDLSC is one of the important cells in the self-defense response to chronic periapical periodontitis. RT-qPCR determined that the level of Klotho expression was increased in $H_2O_2$-induced hPDLSCs senescence. LY294002 is an inhibitor of PI3K. After pretreatment with LY294002, the senescence and apoptosis caused by $H_2O_2$ stimulation in hPDLSCs were reversed. Notably, the decreased Klotho level could be detected, which further confirmed the role of Klotho in the PI3K/AKT/FoxO1 signaling pathway (*Liu et al., 2023*).

Figure 3 is a conclusion of current studied signaling in craniofacial bone remodeling under the regulating of Klotho. Under the condition of inflammation or oxidative stress, Klotho serves as a positive effector of bone regeneration. The signaling pathways modulated by Klotho play crucial roles in promoting bone formation and repair. These consequences in orofacial bone provide valuable insights for further research on long bones.

### Other signaling pathways

In fact, although not extensively studied, some other molecules and pathways could be the downstream of Klotho to affect bone. Mammalian target of rapamycin complex 1 (mTORC1) signaling is overactivated during aging, resulting in a progressive decrease of

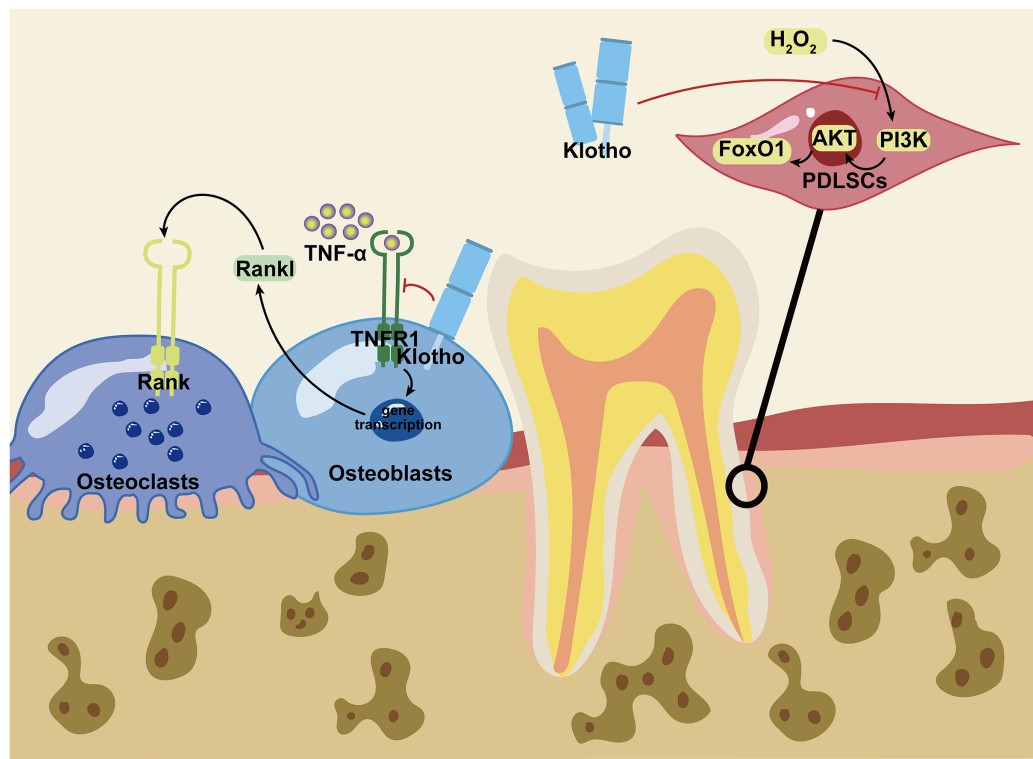

**Figure 3 Mechanism of Klotho in regulating craniofacial bone.** Klotho repressed $H_2O_2$ induced PI3K/AKT/FoxO1 signaling to enhance FoxO1-mediated antioxidant expression and promoted the osteogenesis potential of PDLSCs. Additionally, Klotho inhibited the combination of TNF-α and TNFR1 to limit Rankl secreted by osteoblasts. Therefore, it prevented the binding of Rank and Rankl to restrain osteoclastogenesis under inflammation. PDLSCs, periodontal ligament stem cells. Figure created using Adobe Illustrator.

multidirectional differentiation potential as well as the tissue repair capacity (*Gharibi et al., 2014*). Rapamycin is a mTORC1 inhibitor and autophagy activator. Autophagy activated by rapamycin restored bone mass loss in senescent mice (*Ma et al., 2018*). A significantly higher activity of the downstream effector *i.e.*, 4E-BP1 of mTORC1 was detected in 5 days of primary *kl/kl* mice BMSCs, however, it decreased markedly in 9 days. This illustrated that mTORC1 could be activated in BMSCs during the fast proliferation period, which restrains autophagy and induces stem cells to prepare for entry into the cell cycle (*Rodgers et al., 2014*). Meanwhile, rapamycin increased the number of quiescent BMSCs from *kl/kl* mice, reversed the bone phenotype and extended the life span of mice, indicating that Klotho activated mTOR to weaken autophagy (*Feng et al., 2023*).

UCP2 is a member of the mitochondrial transporter family, which has an inhibitory effect on oxidative stress in cells (*Shen et al., 2024*). Studies have demonstrated that during chronic periodontitis, the mitochondrial pathway takes action on the apoptosis of PDLSCs (*Donadelli et al., 2014*). Klotho can inhibit $H_2O_2$-induced oxidative stress and reduce apoptosis. $H_2O_2$-induced PDLSCs treated with Klotho exhibited increased expression of UCP2 protein. The rescue experiment with si-UCP2 showed that the suppressive impact of Klotho on oxidative stress was significantly weakened in UCP2-silenced cells, which was

manifested by the upregulated oxidative stress marker ROS and the downregulated antioxidant enzymes SOD, GSH-PX and CAT levels compared to the control group. TUNEL assay further found that the downregulation of UCP2 offset the suppression effect of Klotho during cell apoptosis.

## CONCLUSIONS

With the identification of Klotho in bone and the comprehensive investigation of skeletal regeneration, an accumulated number of research concentrated on the function of Klotho in bone development, repair and regeneration emerged. Markedly, the promising role of Klotho in enhancing bone formation under inflammation further confirmed the possible application of Klotho in healing bone defects. Moreover, recent clinical research revealed a negative correlation between serum α-Klotho level and alveolar bone loss (*Wang et al., 2024*). Additionally, the elevated TNF-α level and the enhanced spontaneous osteoclastogenesis is associated with reduced Klotho level (*Salamanna et al., 2024*). These findings strengthened the curative promise of Klotho in bone. Notably, the therapeutic potential of Klotho has been confirmed in non-human primate. For improving cognition, in 2023, Nature Aging published an article illustrating that a single low-dose (10 μg/kg) injection of recombinant Klotho protein can enhance the memory of elderly rhesus (*Castner et al., 2023*). This finding is regarded as further evidence for the potential clinical use of Klotho since primates are more genetically and physiologically similar to human (*Tozer, 2023*).

However, we should be cautious that the use of Klotho for clinical treatment requires careful consideration. First, Klotho has not been utilized in human subjects based on our current knowledge. Secondly, there are no standardized measurement units, different researchers use various units such as ug/kg, pg/ml and ng/mmol. Meanwhile, methods to measure the Klotho levels should be standardized as well. Finally, there is still a lack of research on the appropriate dosage of Klotho. Since Klotho is closely associated with phosphate metabolism, improper application may significantly disrupt mineral homeostasis. Further *in vivo* experiments or clinical research could help to improve the limitations.

Besides, since we already know the role of Klotho in regulating stem cell multidirectional differentiation and promoting bone repairing. With the ability to treat certain primary diseases that could cause bone defects such as inflammation, cancer and mineral disorders (*Kaszubowska et al., 2023*; *Kuro-o, 2011*; *Sariboyaci et al., 2022*). Future directions may focus on the therapeutic effect of Klotho protein in bone defects. Moreover, inflammation is an ordinary pathological process in many diseases. The role of Klotho in anti-inflammation is one of the foundations of treating disease. However, the related study is inadequate and further research is needed. As an exemplification, since TNF-α might directly promote osteogenesis of macrophages, cells in the inflammatory area might improve Rankl response. Further studies could focus on the relationship between mesenchymal progenitor-Klotho and immune cells (*Fan et al., 2022*).

### Funding

This work was supported by grants from the National Natural Science Foundation of China (NSFC) (82370945, 82171001 and 82222015); the State Key Laboratory of Oral Diseases Open Funding Grant Sichuan University (SKLOD-R016); the Research Funding from West China School/Hospital of Stomatology, Sichuan University (RCDWJS2024-(4)); and the Natural Science Foundation of Sichuan Province (2024NSFSC0545). The funders had no role in study design, data collection and analysis, decision to publish, or preparation of the manuscript.

### Grant Disclosures

The following grant information was disclosed by the authors:
National Natural Science Foundation of China (NSFC): 82370945, 82171001 and 82222015.
State Key Laboratory of Oral Diseases Open Funding Grant Sichuan University: SKLOD-R016.
Research Funding from West China School/Hospital of Stomatology, Sichuan University: RCDWJS2024-(4).
Natural Science Foundation of Sichuan Province: 2024NSFSC0545.

### Competing Interests

The authors declare that they have no competing interests.

### Author Contributions

- Xinyu Chen analyzed the data, prepared figures and/or tables, authored or reviewed drafts of the article, and approved the final draft.
- Yali Wei analyzed the data, prepared figures and/or tables, authored or reviewed drafts of the article, and approved the final draft.
- Zucen Li analyzed the data, prepared figures and/or tables, authored or reviewed drafts of the article, and approved the final draft.
- Chenchen Zhou analyzed the data, authored or reviewed drafts of the article, and approved the final draft.
- Yi Fan analyzed the data, authored or reviewed drafts of the article, and approved the final draft.

### Data Availability

This is a literature review.

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
