# Peer review of "Distinct role of Klotho in long bone and craniofacial bone: skeletal development, repair and regeneration"

_PeerJ, doi:10.7717/peerj.18269_

## Round 0.1 · original submission · Minor Revisions

Dear Dr. Zhou,

Your manuscript titled " Distinct role of Klotho in long bone and craniofacial bone: Skeletal development, repair and regeneration" was considered by two expert reviewers and based on their opinions and my review, the decision is “Minor Revisions”.

Please carefully read the reviewers’ comments and address them fully in your revised manuscript. In addition, please address the following points:

(1) When the authors discuss signaling (PI3K/AKT, TNFR, FGF-23, NF-κB and Wnt) they almost solely focus on bone remodeling, but what about bone modeling (bone modeling is only mentioned in L571 (Wnt))? Are all other signaling molecules affect only bone remodeling.
(2) Legends to figures 2 and 3 are very minimal and do not describe the figures in detail.
(3) Conclusions section (L680-699): This does not read like conclusions. The authors should move most of the info to the intro as it supplies info the reader would like to know when they start reading the review (e.g., what is klotho). Furthermore, Klotho-derived peptides (1, 2 and 6) were never mentioned in the text (they do appear in figure 1 with no explanation). They should not appear for the first time in the conclusions section.

- L77: Figure 1 legend. If the review focuses solely on alpha- Klotho, why does the figure also show beta- Klotho and gamma- Klotho? Also, please also explain 12aa, KP1, 2 and 6 in the legend
- L600: “more strong” should be “stronger”.
- L606: Typo. “reporteds” should be “reported”,

Please ensure that all review, editorial, and staff comments are addressed in a response letter and any edits or clarifications mentioned in the letter are also inserted into the revised manuscript where appropriate.

Please note that submitting a revision of your manuscript does not guarantee eventual acceptance, and that your revision may be subject to re-review by the reviewer(s) before a decision is rendered.

Reviewer 1 ·

Basic reporting

The review provides a thorough analysis of the multifaceted roles of the Klotho protein in bone biology. The authors articulate how Klotho, a pivotal protein in aging and mineral ion homeostasis, exhibits differential functions in long bones and craniofacial bones due to their distinct embryonic origins, ossification processes, and cellular compositions. The review underscores the potential of Klotho in clinical applications for bone repair and regeneration, focusing on its involvement in various cellular mechanisms and signaling pathways. However, several concerns are necessary to enhance the manuscript.

Experimental design

1. Emphasize the existing gaps in the literature and clearly outline the unique contributions of this review in the introduction. Contextualize the importance of Klotho in skeletal biology and its clinical implications.
2. Incorporate more recent and seminal studies to provide a balanced and comprehensive perspective. Analyze conflicting findings with nuance, addressing potential reasons and implications for future research.
3. Improve the manuscript's coherence and logical structure with clearly defined subsections. Use explicit headings and subheadings to facilitate easier navigation and enhance overall readability.
4. Seamlessly integrate figures and tables into the text by explicitly referencing them in relevant sections and explaining their significance and contribution to the overall narrative.

Validity of the findings

5. Integrate the various signaling pathways involving Klotho into a cohesive narrative, linking them to Klotho's role in bone biology to provide clearer mechanistic insights. Elucidate how these pathways interact and contribute to phenotypic outcomes.
6. Enhance comprehension by providing a detailed description of the methodologies used in the studies discussed, facilitating a better understanding of the findings and conclusions.
7. Improve the conclusion by offering a concise summary of key findings and a detailed discussion of future research directions. Highlight unresolved questions and propose potential experimental approaches to add value.

Additional comments

8. A deeper exploration of translating Klotho research into clinical practice is needed, including a critical evaluation of potential therapeutic strategies, their feasibility, and challenges. Recent clinical studies on Klotho in craniofacial and alveolar bone, such as DOI: 10.1093/gerona/glae172, could be reviewed to enhance the manuscript's clinical significance. Incorporating these findings will bolster its relevance and applicability to clinical practice.

·

Basic reporting

Generally, it is a well written paper, with sufficient literature references, although the data regarding Klotho protein and its mechanism of action remains scarce.
Well structured with well designed figures and tables.
Only minor revisions needed.

Experimental design

The study is well designed and in my opinion especially in the signaling pathways section, is till now probably the most detailed review in literature
Also the figures are well designed
I would like to stress certain points, requiring form the side of the authors minor revisions:
1) In the section of long bone more emphasis should be implied on the non well understood mechanisms of klotho impact on bone remodelling and mineral homeostasis, which explains the presence of conflicting results on various studies
2) In the section of craniofacial bones, it is very interesting the assumption, that lower klotho levels lead to elevated bone resorption in inflammatory states, but it is demonstrated only in one study, it should be given more emphasis on the fact that these findings need further validation.
3) Concerning the possible therapeutic applications of klotho protein administration, I believe at this point we should be very cautious for the following reasons:
No klotho protein has up to our current knowledge being administered to humans (experimentally has been administered only in mice and non-human primates). Correctly, the authors have suggested the possibility of future clinical applications of Klotho protein, but in my opinion they should make a comment that there is no standardized method regarding the measurement units (μg/kg,pg/ml, ng/m…) and no knowledge about the necessary dose (as the authors mention excess of klotho is related with phosphorus imbalance).

Validity of the findings

Conclusions are well stated, always linked to original research questions
The most interesting argument is the statement about future therapeutic implications of Klotho protein regarding its healing and antiinflammatory properties (according to preclinical data)

Additional comments

-

---

## Round 0.2 · accepted · Accept

Dear Dr. Zhou,

After careful consideration of your revisions and the reviewer's comments, we have determined that your work meets our publication standards. Thus, I am pleased to inform you that your revised manuscript, "Distinct role of Klotho in long bone and craniofacial bone: Skeletal development, repair and regeneration", has been accepted for publication.

Reviewer 1 ·

Basic reporting

no comment

Experimental design

no comment

Validity of the findings

no comment

Additional comments

The authors have addressed all concerns and I believe the manuscript have met the criteria for publication.

·

Basic reporting

As I mentioned in my review,it is a well written paper, with sufficient literature references, although the data regarding Klotho protein and its mechanism of action remains scarce.
The authors replied in all questions posed by the reviewers.

Experimental design

no comment

Validity of the findings

the supported argument is well developed and the authors made all the necessary clarifications

Additional comments

no additional comments